# Efficient Predictive Counterfactual Regret Minimization$^+$ Algorithm in Solving Extensive-Form Games

## Abstract

Imperfect-information extensive-form games (IIGs) serve as a foundational model for capturing interactions among multiple agents in sequential settings with hidden information. A common objective of IIGs is to calculate a Nash equilibrium (NE). Counterfactual Regret Minimization (CFR) algorithms have been widely developed to learn an NE in two-player zero-sum IIGs. Among CFR algorithms, Predictive CFR$^+$ (PCFR$^+$) is powerful, usually achieving an extremely fast empirical convergence rate. However, PCFR$^+$ suffers from the significant discrepancy between strategies represented by explicit accumulated counterfactual regrets across two consecutive iterations, which decreases the empirical convergence rate of PCFR$^+$ in practice. To mitigate this significant discrepancy, we introduce a novel and effective variant of PCFR$^+$, termed Pessimistic PCFR$^+$ (P2PCFR$^+$), minimizing the discrepancy between strategies represented by implicit and explicit accumulated regrets within the same iteration. We provide theoretical proof to show that P2PCFR$^+$ exhibits a faster theoretical convergence rate than PCFR$^+$. Experimental results demonstrate that P2PCFR$^+$ outperforms other tested CFR variants.

## 1 Introduction

Imperfect-information extensive-form games (IIGs) are foundational models to capture interactions among multiple agents in sequential settings with hidden information. IIGs are widely used to simulate real-world scenarios such as medical treatment (Sandholm, 2015), security games (Lisỳ et al., 2016), cybersecurity (Chen et al., 2017), and recreational games (Brown & Sandholm, 2018; 2019b). A primary goal in addressing IIGs is to compute a Nash equilibrium (NE), which represents a rational state where no player can unilaterally improve its payoff by deviating from the equilibrium.

As with much of the literature on solving IIGs, we focus on learning an NE in two-player zero-sum IIGs. The most widely used method for learning an NE in two-player zero-sum IIGs is Counterfactual Regret Minimization (CFR) (Shalev-Shwartz & Singer, 2007; Lanctot et al., 2009; Tammelin, 2014; Brown & Sandholm, 2019a; Farina et al., 2021; 2019; Liu et al., 2021; Pérolat et al., 2021; Liu et al., 2023; Meng et al., 2023; Farina et al., 2023; Xu et al., 2022; 2024b;a; Zhang et al., 2024), as evidenced by their success in superhuman game AIs (Bowling et al., 2015; Moravčík et al., 2017; Brown & Sandholm, 2018; 2019b; Pérolat et al., 2022). CFR algorithms decompose the total regret over the game into a sum of counterfactual regrets associated within information sets (infosets) and employ a local regret minimizer to minimize counterfactual regrets within each infoset.

Many technologies have been proposed to improve the empirical convergence rate of CFR algorithms. For instance, Counterfactual Regret Minimization$^+$ (CFR$^+$) (Tammelin, 2014) replaces the local regret minimizer—Regret Matching (RM) (Hart & Mas-Colell, 2000; Gordon, 2006)—used in vanilla CFR (Zinkevich et al., 2007) with Regret Matching$^+$ (RM$^+$). RM$^+$ improves the empirical convergence rate by ensuring that the accumulated counterfactual regrets remain non-negative. Subsequently, Farina et al. (2021) introduce Predictive CFR$^+$ (PCFR$^+$), an improved variant of CFR$^+$, which leverages the key insight of looking one step ahead to improve the convergence rate. Specifically, PCFR$^+$ maintains two types of accumulated counterfactual regrets: the implicit and the explicit. The phrase "looking one step ahead" refers to PCFR$^+$ making a prediction at each iteration $t$ and using this prediction to derive new explicit accumulated counterfactual regrets from the

current implicit counterfactual regret. PCFR$^+$ then observes the instantaneous counterfactual regret by following the strategy defined by the new explicit accumulated counterfactual regret and uses it to update the current implicit counterfactual regret. PCFR$^+$ sets the instantaneous counterfactual regret observed at iteration $t-1$ as the prediction at iteration $t$. If the prediction aligns with the observed instantaneous counterfactual regret, the theoretical convergence rate of PCFR$^+$ can be improved improves from $O(1/\sqrt{T})$ of CFR$^+$ to $O(1/T)$ (Farina et al., 2021).

Unfortunately, due to the significant discrepancy between strategies represented by explicit accumulated counterfactual regrets across two consecutive iterations, the alignment between the prediction and with the observed instantaneous counterfactual regret often fails, which reduces empirical convergence rate. For instance, as illustrated in Section 4.1, within a given infoset, such strategies may be $[1;0]$ and $[0;1]$, respectively. Such a significant discrepancy indicates that the prediction at $t$, *i.e.*, instantaneous counterfactual regret observed at iteration $t-1$, may be completely different from the instantaneous counterfactual regret observed at iteration $t$ (Farina et al., 2023). It implies that this alignment fails, which undermines the empirical convergence rate of PCFR$^+$, as evidenced by our experiments utilizing the open-source implementation of CFR algorithms (Liu et al., 2024)[1]. Precisely, PCFR$^+$ converges more slowly than other classical CFR algorithms, such as CFR$^+$, even in standard IIG benchmarks like Leduc Poker, Goofspiel, and Liar's Dice.

To mitigate the discrepancy between strategies represented by explicit accumulated counterfactual regrets across two consecutive iterations, we propose a novel variant of PCFR$^+$, named Pessimistic PCFR$^+$ (P2PCFR$^+$). The key insight of P2PCFR$^+$ is to reduce the discrepancy between strategies represented by implicit and explicit accumulated regrets within the same iteration to diminish the significant discrepancy between strategies represented by explicit accumulated counterfactual regrets across two consecutive iterations. Specifically, from the Cauchy-Schwarz inequality (Steele, 2004), the discrepancy between strategies represented by explicit accumulated regrets at iteration $t$ and $t+1$ is bounded by the sum of (i) the discrepancy between strategies represented by the implicit and explicit accumulated regrets at iteration $t$, (ii) the discrepancy between strategies represented by the implicit and explicit accumulated regrets at iteration $t+1$, and (iii) the discrepancy between strategies represented by the explicit accumulated regrets at iteration $t$ and $t+1$. Obviously, the discrepancy between strategies represented by the explicit accumulated counterfactual regrets at iterations $t$ and $t+1$ depends largely on the discrepancy between the strategies represented by the implicit and explicit accumulated counterfactual regrets within the same iteration. As the latter discrepancy decreases, the former discrepancy also reduces, which accelerates the empirical convergence rate.

We show that P2PCFR$^+$ enjoys a lower regret bound than PCFR$^+$. In other words, P2PCFR$^+$ exhibits a faster theoretical convergence rate than PCFR$^+$. Moreover, the implementation of P2PCFR$^+$ is also notably straightforward, requiring only a single line adjustment compared to the open-source implementation of PCFR$^+$ (Liu et al., 2024).

We conduct extensive experimental evaluations of P2PCFR$^+$ across nine instances from four standard IIG benchmarks: Kuhn Poker, Leduc Poker, Goofspiel, and Liar's Dice, compared with previous CFR algrithms. Among the tested algorithms, P2PCFR$^+$ achieves the fastest empirical convergence rate. Furthermore, our experiments reveal that the improvement of P2PCFR$^+$ over PCFR$^+$ in terms of empirical convergence rate is highly correlated with the reduction of the discrepancy between the strategies represented by implicit and explicit accumulated counterfactual regrets, relative to PCFR$^+$.

Concretely, we make the following contributions:

- We introduce a novel and effective variant of PCFR$^+$, called P2PCFR$^+$. P2PCFR$^+$ mitigates the discrepancy between strategies represented by explicit accumulated counterfactual regrets across two consecutive iterations by reducing the discrepancy between strategies represented by implicit and explicit accumulated regrets within the same iteration.

- We prove that P2PCFR$^+$ exhibits faster theoretical convergence rate than PCFR$^+$.

- We demonstrate that P2PCFR$^+$ is implementable with a single-line modification to the open-source PCFR$^+$ implementation.

- Experimental results from four standard IIG benchmarks demonstrate that P2PCFR$^+$ outperforms other tested CFR variants.

---

[1]https://github.com/liumy2010/LiteEFG

## 2 RELATED WORK

We focus on CFR algorithms (Shalev-Shwartz & Singer, 2007; Lanctot et al., 2009; Tammelin, 2014; Brown & Sandholm, 2019a; Farina et al., 2021; 2019; Liu et al., 2021; Pérolat et al., 2021; Liu et al., 2023; Meng et al., 2023; Farina et al., 2023; Xu et al., 2022; 2024b;a; Zhang et al., 2024), the most widely used method for learning an NE in two-player zero-sum IIGs, as evidenced by their success in superhuman game AIs (Bowling et al., 2015; Moravčík et al., 2017; Brown & Sandholm, 2018; 2019b; Pérolat et al., 2022).

The key insight of CFR algorithms is the decomposition of the regret over the game into the sum of counterfactual regrets associated within infosets. The vanilla CFR algorithm, introduced by Shalev-Shwartz & Singer (2007), employs RM (Hart & Mas-Colell, 2000; Gordon, 2006) as the local regret minimizer. To improve the empirical convergence rate of CFR, it is common to design more effective local regret minimizers, as the selection of the local regret minimizers has a significant impact on the overall performance of the CFR algorithm. Examples include RM$^+$ (Bowling et al., 2015), Discounted RM (DRM) (Brown & Sandholm, 2019a), and PRM$^+$ (Farina et al., 2021), which correspond to CFR$^+$ (Bowling et al., 2015), DCFR (Brown & Sandholm, 2019a), and PCFR$^+$ (Farina et al., 2021), respectively. PCFR$^+$ can demonstrate faster empirical convergence than other CFR variants. However, as shown in our experiments, PCFR$^+$ is sometimes outperformed by CFR$^+$ and DCFR, even on standard IIG benchmarks.

To accelerate the convergence rate of PCFR$^+$, several algorithms have been proposed (Farina et al., 2023; Xu et al., 2024a;b; Zhang et al., 2024). For instance, to mitigate the slow empirical convergence rate caused by the discrepancy between strategies represented by explicit accumulated counterfactual regrets, Farina et al. (2023) ensure that the lower bound of the 1-norm of implicit and explicit accumulated counterfactual regrets exceeds a positive constant. However, their algorithms, Stable PCFR$^+$ and Smooth PCFR$^+$, forfeit a crucial property of PCFR$^+$—parameter-free—meaning no parameters need to be tuned (Grand-Clément & Kroer, 2021) to guarantee convergence. P2PCFR$^+$ offers a simpler and more effective solution to address this discrepancy while still obtaining the parameter-free property. In our experiments, we observe that the empirical convergence rate of P2PCFR$^+$ consistently significantly outperforms that of Stable PCFR$^+$ and Smooth PCFR$^+$.

## 3 PRELIMINARIES

**Imperfect-information Extensive-form games (IIGs).** To model tree-form sequential decision-making problems with hidden information, a common used model is IIG (Osborne et al., 2004). An IIG can be formulated as $G = \{\mathcal{N}, \mathcal{H}, P, A, \mathcal{I}, \{u_i\}\}$. Here, $\mathcal{N} = \{0, 1\}$ is the set of players. "Nature" is also considered a player $c$ (representing chance) and chooses actions with a fixed known probability distribution. $\mathcal{H}$ is the set of all possible history sequences. The set of leaf nodes is denoted by $\mathcal{Z}$. For each history $h \in \mathcal{H}$, the function $P(h)$ represents the player acting at node $h$, and $A(h)$ denotes the actions available at node $h$. To account for private information, the nodes for each player $i$ are partitioned into a collection $\mathcal{I}_i$, referred to as information sets (infosets). For any infoset $I \in \mathcal{I}_i$, histories $h, h' \in I$ are indistinguishable to player $i$. The notation $\mathcal{I}$ denotes $\mathcal{I} = \{\mathcal{I}_i | i \in \mathcal{N}\}$. Thus, we have $P(I) = P(h), A(I) = A(h), \forall h \in I$. For each leaf node $z$, there is a pair $(u_0(z), u_1(z)) \in [-1, \ 1]$ which denotes the payoffs for the min player (player 0) and the max player (player 1) respectively. In two-player zero-sum IIGs, $u_0(z) = -u_1(z), \forall z \in \mathcal{Z}$.

**Behavioral strategy.** In this paper, we present the strategy via behavioral strategy. This strategy $\sigma_i$ is defined on each infoset. For any infoset $I \in \mathcal{I}_i$, the probability for the action $a \in A(I)$ is denoted by $\sigma_i(I, a)$. We use $\sigma_i(I) = [\sigma_i(I, a) | a \in A(I)] \in \Delta^{|A(I)|}$ to denote the strategy at infoset $I$, where $\Delta^{|A(I)|}$ is a $(|A(I)| - 1)$-dimension simplex. If every player follows the strategy profile $\sigma = [\sigma_0; \sigma_1]$ and reaches infoset $I$, the reaching probability is denoted by $\pi^\sigma(I)$. The contribution of $i$ to this probability is $\pi_i^\sigma(I)$ and $\pi_{-i}^\sigma(I)$ for other than $i$, where $-i$ denotes the players other than $i$. In IIGs, $u_i(\sigma_i, \sigma_{-i}) = \sum_{z \in \mathcal{Z}} u_i(z) \pi^\sigma(z)$.

**Nash equilibrium (NE).** NE denotes a rational behavior where no player can benefit by unilaterally deviating from the equilibrium. For any player, her strategy is the best-response to the strategies of others. Formally, for all NE strategy profile $\sigma^*$ and $i \in \mathcal{N}$, it holds that $u(\sigma_i^*, \sigma_{-i}^*) \geq u(\sigma_i, \sigma_{-i}^*)$ for all $\sigma$. A widely used metric to measure the distance from the given strategy profile $x$ to NE is exploitability, which is defined as $\epsilon(\sigma) = \sum_{i \in \mathcal{N}} \max_{\sigma_i'} (u(\sigma_i, \sigma_{-i}) - u(\sigma_i', \sigma_{-i}))/|\mathcal{N}|$. If $\epsilon(\sigma) = \delta$, then $\sigma$ is called as a $\delta$-NE.

**learning an NE via regret minimization algorithms.** To learn NE in IIGs, a common used method is regret minimization algorithms (Rakhlin & Sridharan, 2013a;b; Hazan et al., 2016; Joulani et al., 2017). For any sequence of strategies $\sigma_i^1, \cdots, \sigma_i^T$ of player $i$, player $i$'s regret for a fixed strategy $\sigma_i$ is $R_i^T = \sum_{t=1}^{T} u_i(\sigma_i^t, \sigma_{-i}^t) - \min_{\sigma_i} u_i(\sigma_i, \sigma_{-i}^t)$ for all sequence $\sigma_{-i}^1, \cdots, \sigma_{-i}^T$. Regret minimization algorithms are algorithms that ensure $R_i^T$ grows sublinearly from any $\sigma_i$. If each player follow a regret minimization algorithm, then their average strategy converges to NE in two-player zero-sum IIGs. Formally, assume the regret of each player $i$ is $R_i^T$, then it hods that

$$\epsilon(\bar{\sigma}) = \epsilon(\bar{\sigma}_0, \bar{\sigma}_1) \leq \frac{\sum_{i \in \mathcal{N}} R_i^T}{|\mathcal{N}|T}, \tag{1}$$

where $\bar{\sigma}_i(I) = \sum_{t=1}^{T} \pi_i^{\sigma^t}(I)\sigma_i^t(I) / \sum_{t=1}^{T} \pi_i^{\sigma^t}(I)$ for all $i \in \mathcal{N}$ and $I \in \mathcal{I}_i$.

**Counterfactual Regret Minimization (CFR) framework.** This framework (Zinkevich et al., 2007; Farina et al., 2019; Liu et al., 2021; Farina et al., 2023) is designed to learn NE of two-player zero-sum IIGs. Instead of directly minimizing the global regret, this framework decomposes the regret to each infoset and independently minimizes the local regret within each infoset. This framework has facilitated the development of several superhuman game AIs (Bowling et al., 2015; Moravčík et al., 2017; Brown & Sandholm, 2018; 2019b; Pérolat et al., 2022). CFR algorithms are the algorithms which utilize this framework to decompose the regret to each infoset and employ regret minimization algorithms as the local regret minimizers to minimize the regret at each infoset. Let $\sigma^t$ be the strategy profile at iteration $t$. CFR algorithms compute the counterfactual value at infoset $I$ for action $a$ as

$$v^{\sigma^t}(Ia) = \sum_{h \in I} \sum_{z \in \mathcal{Z}_{ha}} \pi_{-i}^{\sigma^t}(h)\pi^{\sigma^t}(ha, z)u_i(z),$$

where $\pi^{\sigma^t}(ha, z)$ denotes the probability from $ha$ to $z$ if all players play according to $\sigma^t$ and $\mathcal{Z}_{ha}$ is the set of the leaf nodes that are reachable after choosing action $a$ at history $h$. For any infoset $I$, the counterfactual regret is

$$R^T(I) = \sum_{t=1}^{T} \sum_{a \in A(I)} \sigma_i^t(Ia)v^{\sigma^t}(Ia) - \min_{a \in A(I)} \sum_{t=1}^{T} v^{\sigma^t}(Ia).$$

It has been shown that the regret over the game $R_i^T = \sum_{t=1}^{T} u_i(\sigma_i^t, \sigma_{-i}^t) - \min_{\sigma_i} u_i(\sigma_i, \sigma_{-i}^t)$ less the sum of the counterfactual regrets within infosets (Zinkevich et al., 2007). Formally,

$$R_i^T \leq \sum_{I \in \mathcal{I}_i} R^T(I). \tag{2}$$

So any regret minimization can be used as the local regret minimizer to minimize the regret $R^T(I)$ over each infoset to minimize the regret $R_i^T$.

**Vanilla Counterfactual Regret Minimization (Vanilla CFR).** The first CFR algorithm is proposed by Zinkevich et al. (2007), which uses Regret Matching (RM) (Hart & Mas-Colell, 2000; Gordon, 2006) as the local regret minimizer. Formally, at each iteration $t$, vanilla CFR updates its accumulated counterfactual regret $\boldsymbol{R}_I^t$ at infoset $I$ according to

$$\boldsymbol{R}_I^{t+1} = \boldsymbol{R}_I^t + \boldsymbol{r}_I^t,$$

where $\boldsymbol{r}_I^t = \langle v^{\sigma^t}(I), \sigma_i^t(I) \rangle \mathbf{1} - v^{\sigma^t}(I)$ is the instantaneous counterfactual regret. Then, vanilla CFR gets new strategies via the regret-matching operator

$$\sigma_i^{t+1}(I) = \frac{[\boldsymbol{R}_I^{t+1}]^+}{\|[\boldsymbol{R}_I^{t+1}]^+\|_1},$$

where $i = P(I)$ and $[\cdot]^+ = \max(\cdot, \mathbf{0})$.

**Counterfactual Regret Minimization$^+$ (CFR$^+$).** To improve the empirical convergence rate of vanilla CFR, Tammelin (2014) propose a variant of vanilla CFR called CFR$^+$, which utilizes Regret Matching$^+$ (RM$^+$) (Tammelin, 2014) as the local regret minimizer. The key insight of RM$^+$ is to ensure that $\boldsymbol{R}_I^t \geq \mathbf{0}$ at each iteration $t$ and infoset $I$. Formally, at each iteration $t$, CFR$^+$ updates its accumulated counterfactual regret $\boldsymbol{R}_I^t$ at infoset $I$ according to

$$\boldsymbol{R}_I^{t+1} = [\boldsymbol{R}_I^t + \boldsymbol{r}_I^t]^+.$$

Then, as did in CFR, CFR$^+$ gets strategies via the regret-matching operator

$$\sigma_i^{t+1}(I) = \frac{[\boldsymbol{R}_I^{t+1}]^+}{\|[\boldsymbol{R}_I^{t+1}]^+\|_1} = \frac{\boldsymbol{R}_I^{t+1}}{\|\boldsymbol{R}_I^{t+1}\|_1},$$

where $i = P(I)$, and the second equality comes from the fact that $\boldsymbol{R}_I^{t+1} \geq \boldsymbol{0}$.

**Predictive Counterfactual Regret Minimization$^+$ (PCFR$^+$).** In this paper, we focus on an advanced variant of CFR$^+$, known as PCFR$^+$ (Farina et al., 2021), which can exhibit a significantly faster empirical convergence rate compared to CFR$^+$. PCFR$^+$ employs Predictive RM$^+$ (PRM$^+$) (Farina et al., 2021) as its local regret minimizer, with its key insight being looking one step ahead, a similar idea to the momentum algorithms in the optimization area (Hoda et al., 2010; Kingma & Ba, 2014). Specifically, at each iteration, PCFR$^+$ maintains two types of accumulated counterfactual regrets: implicit and explicit. The phrase "looking one step ahead" means that PCFR$^+$ makes a prediction and uses this prediction to derive new explicit accumulated counterfactual regrets from the current implicit counterfactual regret. Then, PCFR$^+$ observes the instantaneous counterfactual regret by following the strategy defined by the explicit accumulated counterfactual regret. Lastly, this instantaneous counterfactual regret is subsequently used to update the current implicit counterfactual regret. If the prediction aligns with the observed instantaneous counterfactual regret, Farina et al. (2021) show that the theoretical convergence of PCFR$^+$ can be improved from $O(1/\sqrt{T})$ of CFR$^+$ to $O(1/T)$. In practice, PCFR$^+$ employs the instantaneous counterfactual regret observed at the last iteration $t - 1$ as the prediction at iteration $t$ to look one step ahead. Formally, at each iteration $t$ and for each infoset $I \in \mathcal{I}$, PCFR$^+$ updates its strategy according to

$$\hat{\boldsymbol{R}}_I^t = [\boldsymbol{R}_I^t + \boldsymbol{r}_I^{t-1}]^+, \boldsymbol{R}_I^{t+1} = [\boldsymbol{R}_I^t + \boldsymbol{r}_I^t]^+,$$

$$\sigma_i^t(I) = \frac{[\hat{\boldsymbol{R}}_I^t]^+}{\|[\hat{\boldsymbol{R}}_I^t]^+\|_1} = \frac{\hat{\boldsymbol{R}}_I^t}{\|\hat{\boldsymbol{R}}_I^t\|_1}, \tag{3}$$

where $i = P(I)$, $\boldsymbol{R}_I^1 = \boldsymbol{0}$, the second equality of the second line comes from the fact that $\hat{\boldsymbol{R}}_I^{t+1} \geq \boldsymbol{0}$, as well as $\boldsymbol{R}_I^{t+1}$ and $\hat{\boldsymbol{R}}_I^{t+1}$ are implicit and explicit accumulated counterfactual regrets, respectively.

## 4 OUR METHOD

Although PCFR$^+$ utilizes prediction to substantially accelerate the empirical convergence rate, it suffers from a significant significant discrepancy between strategies represented by explicit accumulated counterfactual regrets. This discrepancy reduces the empirical convergence rate. To mitigate this issue, we introduce a novel variant of PCFR$^+$, called Pessimistic PCFR$^+$ (P2PCFR$^+$).

### 4.1 EXAMPLE OF SIGNIFICANT DISCREPANCY BETWEEN STRATEGIES

Firstly, we present an example illustrating the substantial discrepancy between strategies represented by explicit accumulated counterfactual regrets in PCFR$^+$. Consider a scenario where the number of available actions for player $i$ at an infoset $I \in \mathcal{I}_i$ is 2. Assume that, at iteration $t$, the implicit accumulated counterfactual regret $\boldsymbol{R}_I^t$ is $[2; 0]$, and the instantaneous counterfactual regret $\boldsymbol{r}_I^{t-1}$ is $[1; -1]$. From the update rule of PCFR$^+$ defined in Eq. (3), the new explicit accumulated counterfactual regret $\hat{\boldsymbol{R}}_I^t$ is $[3; 0]$. Now, assume the observed instantaneous counterfactual regret $\boldsymbol{r}_I^t$ is $[-1; 1]$. Similarly, we derive that $\boldsymbol{R}_I^{t+1}$ and $\hat{\boldsymbol{R}}_I^{t+1}$ are $[1; 1]$ and $[0; 2]$, respectively. Applying the regret matching operator, the strategies represented by $\hat{\boldsymbol{R}}_I^t$ and $\hat{\boldsymbol{R}}_I^{t+1}$ are $[1; 0]$ and $[0; 1]$ (derived from $[3; 0]$ and $[0; 2]$), respectively. This demonstrates a significant discrepancy between strategies ($[1; 0]$ and $[0; 1]$) represented by explicit accumulated counterfactual regrets. From the analysis in Farina et al. (2023), we have

$$\|\boldsymbol{r}_i^{t+1}(I) - \boldsymbol{r}_i^t(I)\|_2^2 \leq O(\sum_{i \in \mathcal{N}} \sum_{I \in \mathcal{I}_i} \|\sigma_i^{t+1}(I) - \sigma_i^t(I)\|_2^2). \tag{4}$$

Therefore, as the discrepancy between strategies represented by explicit accumulated counterfactual regrets is large, the prediction is entirely incorrect, leading to the failure of the alignment between the prediction and with the observed instantaneous counterfactual regret. This failure ultimately impairs the empirical convergence rate of PCFR$^+$, as demonstrated by our experiments using the open-source implementation of CFR algorithms (Liu et al., 2024). Specifically, PCFR$^+$ converges more slowly than classical CFR algorithms, such as CFR$^+$, even in standard IIG benchmarks like Leduc Poker, Goofspiel, and Liar's Dice.

## 4.2 PESSIMISTIC PCFR$^+$

To address the discrepancy between strategies represented by explicit accumulated counterfactual regrets across two consecutive iterations, we propose a novel variant of PCFR$^+$, termed Pessimistic PCFR$^+$ (P2PCFR$^+$). P2PCFR$^+$ leverages the misalignment of step sizes in the updates of implicit and explicit accumulated counterfactual regrets to make more conservative predictions. This approach reduces the discrepancy between strategies represented by these two types of regrets within the same iteration, leading to a lower overall discrepancy in strategies across consecutive iterations.

Specifically, assume the strategy represented by $\boldsymbol{R}_I^t$ is $\tilde{\sigma}_i^t(I) = \boldsymbol{R}_I^t / \|\boldsymbol{R}_I^t\|_1$. From the Cauchy-Schwarz inequality (Steele, 2004), we have that the discrepancy $\|\sigma_i^{t+1}(I) - \sigma_i^t(I)\|_2^2$ between strategies represented by explicit accumulated regrets at iteration $t$ and $t+1$ is bounded as

$$\|\sigma_i^{t+1}(I) - \sigma_i^t(I)\|_2^2 \le 3\|\sigma_i^{t+1}(I) - \tilde{\sigma}_i^{t+1}(I)\|_2^2 + 3\|\tilde{\sigma}_i^{t+1}(I) - \tilde{\sigma}_i^t(I)\|_2^2 + 3\|\tilde{\sigma}_i^t(I) - \sigma_i^t(I)\|_2^2, \quad (5)$$

where $\|\sigma_i^t(I) - \tilde{\sigma}_i^t(I)\|_2^2$ and $\|\tilde{\sigma}_i^{t+1}(I) - \sigma_i^{t+1}(I)\|_2^2$ are the discrepancy between strategies represented by the implicit and explicit accumulated regrets at iteration $t$ and $t+1$, respectively, as well as $\|\tilde{\sigma}_i^{t+1}(I) - \tilde{\sigma}_i^t(I)\|_2^2$ is the discrepancy between strategies represented by the explicit accumulated regrets at iteration $t$ and $t+1$.

To reduce the value of $\|\sigma_i^{t+1}(I) - \sigma_i^t(I)\|_2^2$, P2PCFR$^+$ exploits the misalignment of step sizes in the updates of implicit and explicit accumulated regrets. This results in a more pessimistic prediction compared to PCFR$^+$, leading to lower values of $\|\sigma_i^t(I) - \tilde{\sigma}_i^t(I)\|_2^2$ and $\|\tilde{\sigma}_i^{t+1}(I) - \sigma_i^{t+1}(I)\|_2^2$, while maintaining the same value of $\|\tilde{\sigma}_i^{t+1}(I) - \tilde{\sigma}_i^t(I)\|_2^2$. Formally, at each iteration $t$, P2PCFR$^+$ updates its strategy at each infoset $I$ according to

$$\hat{\boldsymbol{R}}_I^t = [\boldsymbol{R}_I^t + \frac{1}{1+\alpha} \boldsymbol{r}_I^{t-1}]^+, \boldsymbol{R}_I^{t+1} = [\boldsymbol{R}_I^t + \boldsymbol{r}_I^t]^+,$$

$$\sigma_i^t(I) = \frac{[\hat{\boldsymbol{R}}_I^t]^+}{\|[\hat{\boldsymbol{R}}_I^t]^+\|_1} = \frac{\hat{\boldsymbol{R}}_I^t}{\|\hat{\boldsymbol{R}}_I^t\|_1},$$

where $i = P(I)$ and $\alpha \ge 0$ is a constant. Obviously, the value of $\|\tilde{\sigma}_i^{t+1}(I) - \tilde{\sigma}_i^t(I)\|_2^2$ remains the same as in PCFR$^+$. Furthermore, P2PCFR$^+$ intuitively reduces the gap between implicit and explicit accumulated counterfactual regrets, leading to a smaller discrepancy between $\tilde{\sigma}_i^t(I)$ and $\sigma_i^t(I)$ (as well as between $\tilde{\sigma}_i^{t+1}(I)$ and $\sigma_i^{t+1}(I)$). From Eq. (5), as these discrepancy decreases, the discrepancy $\|\sigma_i^{t+1}(I) - \sigma_i^t(I)\|_2^2$ between the strategies represented by explicit regrets at iterations $t$ and $t+1$ also decreases. We first provide the regret bound when the gap between the prediction and the observed instantaneous counterfactual regret is considered, as shown in Theorem 4.1.

**Theorem 4.1.** *(Proof is in Appendix A) Assume that $T$ iterations of P2PCFR$^+$ with any $1 \ge \alpha \ge 0$ are conducted. Then the counterfactual regret at any infoset $I \in \mathcal{I}$ is bound by*

$$R^T(I) \le 2\sqrt{\frac{2+\alpha}{1+\alpha} EF \sum_{t=1}^T \frac{1}{2}\|\boldsymbol{r}_I^t - \boldsymbol{r}_I^{t-1}\|_2}.$$

*where $E = \max_{\boldsymbol{R}_I \in \Delta^{|A(I)|}} \frac{1}{2}\|\boldsymbol{R}_I - \boldsymbol{R}_I^1\|_2^2$, $F = \max_{I \in \mathcal{I}} \|\boldsymbol{r}_I\|_2$ and $\eta > 0$.*

From the property of the counterfactual regret as shown in Eq. (2), we get

$$R_i^T \le \sum_{I \in \mathcal{I}_i} R^T(I) \le O\left(2|\mathcal{I}_i|\sqrt{\frac{2+\alpha}{1+\alpha} EF \sum_{t=1}^T \frac{1}{2}\|\boldsymbol{r}_I^t - \boldsymbol{r}_I^{t-1}\|_2}\right). \quad (6)$$

Then, from Eq. (1) and (6), we can get the upper bound of the exploitability of the average strategy of P2PCFR$^+$, as shown in Theorem 4.2.

**Theorem 4.2.** *Assume that $T$ iterations of P2PCFR$^+$ with any $1 \ge \alpha \ge 0$ are conducted in a two-player zero-sum IIG. Then the exploitability of the average strategy of P2PCFR$^+$ is bound by*

$$\epsilon(\bar{\sigma}) = \epsilon(\bar{\sigma}_0, \bar{\sigma}_1) \le O\left(2|\mathcal{I}_i|\sqrt{\frac{2+\alpha}{1+\alpha} EF \sum_{t=1}^T \frac{1}{2}\|\boldsymbol{r}_I^t - \boldsymbol{r}_I^{t-1}\|_2}\right),$$

*where $\bar{\sigma}_i(I) = \sum_{t=1}^T \pi_i^{\sigma^t}(I)\sigma_i^t(I) / \sum_{t=1}^T \pi_i^{\sigma^t}(I)$ for all $i \in \mathcal{N}$, and $I \in \mathcal{I}_i$.*

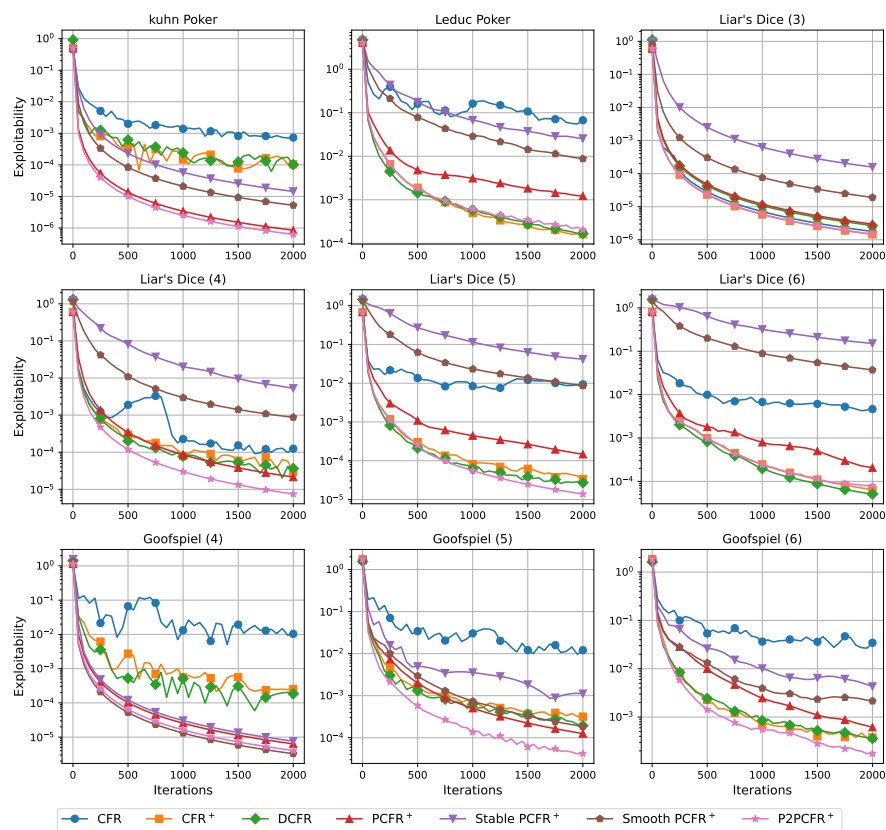

Figure 1: Empirical Convergence results of tested CFR algorithms. Each algorithm runs for 2,000 iterations. In all plots, the x-axis is the number of iteration, and the y-axis represents exploitability, displayed on a logarithmic scale. Liar's Dice ($x$) represents that every player is given a die with $x$ sides. Goofspiel ($x$) denotes that each player is dealt $x$ cards.

As $\alpha$ increases, $(2 + \alpha)/(1 + \alpha)$ decreases. When $\alpha \to 0$, P2PCFR$^+$ simplifies to PCFR$^+$, indicating that P2PCFR$^+$ achieves a faster theoretical convergence rate compared to PCFR$^+$, provided that the value of $\|\boldsymbol{r}_I^t - \boldsymbol{r}_I^{t-1}\|_2^2$ remains the same for both P2PCFR$^+$ and PCFR$^+$. More importantly, since the updates in P2PCFR$^+$ are more pessimistic, $\|\sigma_i^t(I) - \tilde{\sigma}_i^t(I)\|_2^2$ is expected to decrease, which implies a reduction in $\|\sigma_i^{t+1}(I) - \sigma_i^t(I)\|_2^2$ according to Eq. (6). Consequently, from Eq. (4), we deduce that the difference $\|\boldsymbol{r}_i^{t+1}(I) - \boldsymbol{r}_i^t(I)\|_2^2$ in instantaneous counterfactual regret across iterations diminishes. As shown in Theorem 4.2, this reduction leads to a faster convergence rate.

Theorem 4.1 considers the gap between the prediction and the observed instantaneous counterfactual regret. Now, we provide an alternative regret bound as shown in Theorem 4.3, which does not consider this gap. In other words, it considers the worst case.

**Theorem 4.3.** *(Proof is in Appendix B) Assume that $T$ iterations of P2PCFR$^+$ with any $\alpha \geq 0$ are conducted. Then the counterfactual regret at any infoset $I \in \mathcal{I}$ is bound by*

$$R^T(I) \leq O\left(2\sqrt{(1 + \frac{1}{(1+\alpha)^2})EF^2T}\right).$$

**Theorem 4.4.** *Assume that $T$ iterations of P2PCFR$^+$ are conducted in a two-player zero-sum IIG. Then the exploitability of the average strategy of P2PCFR$^+$ is bound by*

$$\epsilon(\bar{\sigma}) = \epsilon(\bar{\sigma}_0, \bar{\sigma}_1) \leq O\left(|\mathcal{I}_i|\sqrt{(1 + \frac{1}{(1+\alpha)^2})\frac{1}{T}}\right),$$

*where $\bar{\sigma}_i(I) = \sum_{t=1}^T \pi_i^{\sigma^t}(I)\sigma_i^t(I)/\sum_{t=1}^T \pi_i^{\sigma^t}(I)$ for all $i \in \mathcal{N}$ and $I \in \mathcal{I}_i$.*

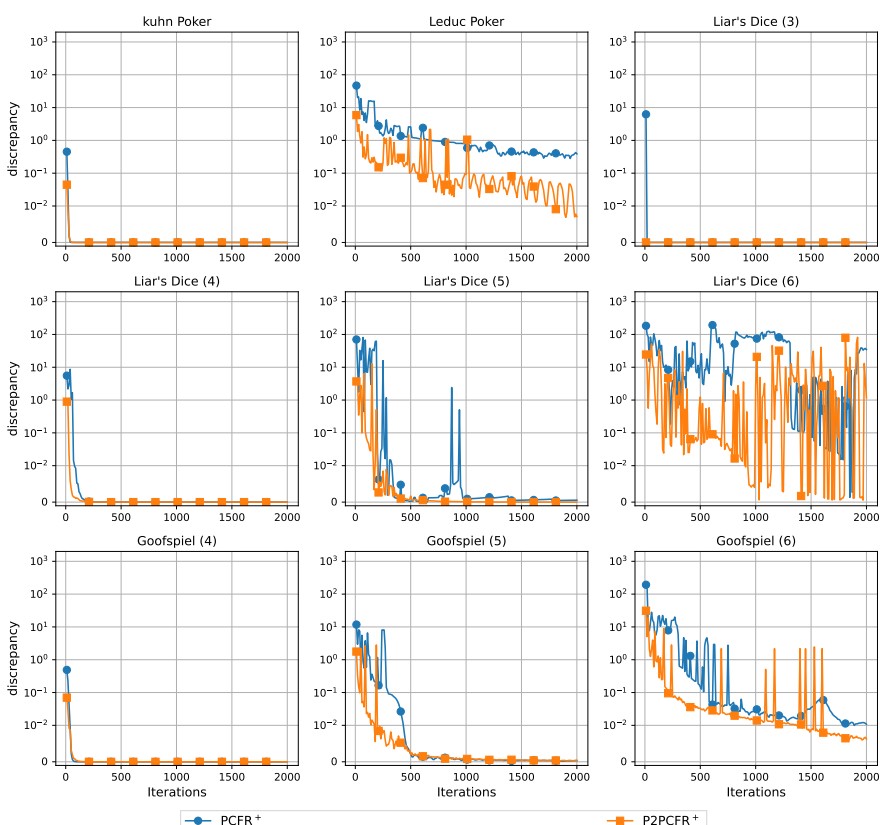

Figure 2: Discrepancy between strategies represented by the implicit and explicit accumulated counterfactual regrets of PCFR$^+$ and P2PCFR$^+$.

Similarly, an alternative upper bound on the exploitability of the average strategy in P2PCFR$^+$ is obtained, as shown in Theorem 4.4. As $\alpha$ increases, $1/(1 + \alpha)^2$ decreases, meaning that increasing $\alpha$ leads to a reduction in $\sqrt{(1 + 1/(1 + \alpha)^2)}$. As $\alpha \to 0$, P2PCFR$^+$ reduces to PCFR$^+$, which means P2PCFR$^+$ achieves faster theoretical convergence rate than PCFR$^+$. As $\alpha \to \infty$, P2PCFR$^+$ reduces to CFR$^+$. While Theorem 4.4 indicates the fastest convergence rate when $\alpha \to \infty$, $\alpha \to \infty$ also implies that the key insight of PCFR$^+$—looking one step ahead—diminishes, leading to a weaker empirical convergence rate. This is verified in our experiments, where CFR$^+$ converges more slowly than APCFR$^+$ in most games.

In conclusion, from Theorem 4.2 and Theorem 4.4, regardless of whether the gap between the prediction and observed instantaneous counterfactual regret is considered, P2PCFR$^+$ exhibits faster theoretical convergence rate than PCFR$^+$.

## 5 EXPERIMENTS

We employ four standard commonly used IIG benchmarks to evaluate the empirical convergence rate of P2PCFR$^+$, *e.g.*, Kuhn Poker, Leduc Poker, Goofspiel Poker, and Liar's Dice. All the tested games are implemented by OpenSpiel (Lanctot et al., 2019). Our evaluation includes a comparison with several existing PCFR$^+$ variants as well as other classical CFR algorithms. Specifically, we compare P2PCFR$^+$ against PCFR$^+$ (Farina et al., 2021), Stable PCFR$^+$ (Farina et al., 2023), Smooth PCFR$^+$ (Farina et al., 2023), vanilla CFR (Zinkevich et al., 2007), CFR$^+$ (Tammelin, 2014), and DCFR (Brown & Sandholm, 2019a). Each algorithm is run for 2000 iterations, which is sufficient to achieve low exploitability for solving the tested games. We employ alternating updates and linear averaging (except for DCF), both of which are standard techniques to improve the empirical convergence rate of CFR algorithms. For P2PCFR$^+$, although Theorem 4.2 requires $\alpha \leq 1$, we

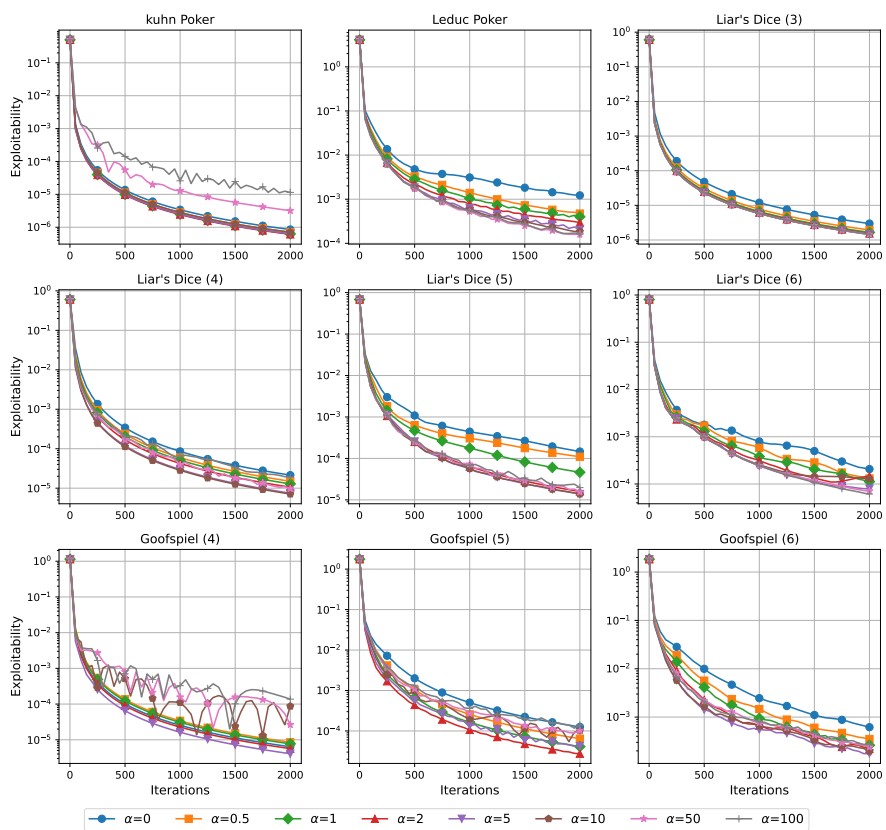

Figure 3: Empirical Convergence results of P2PCFR$^+$ with different $\alpha$. When $\alpha \to 0$, P2PCFR$^+$ reduces to PCFR$^+$.

set $\alpha = 5$ because it empirically achieves a faster convergence rate than $\alpha = 1$, as detailed below. For both Stable PCFR$^+$ and Smooth PCFR$^+$, we set their learning rates to 1, following the original configuration from their original paper. All algorithm implementations are based on the open-source code of LiteEFG (Liu et al., 2024). All experiments are conducted on a machine with a Xeon(R) Gold 6444Y CPU and 256 GB of memory.

The results about convergence rates are shown in Figure 1. P2PCFR$^+$ algorithm exhibits the fastest empirical convergence rate among all tested PCFR$^+$ variants. Specifically, compared to PCFR$^+$, P2PCFR$^+$ achieves significantly faster empirical convergence rate in 7 games except kuhn Poker and Goofspiel (4), while PCFR$^+$ never outperforms P2PCFR$^+$. In comparison with Stable PCFR$^+$ and Smooth PCFR$^+$, although these two PCFR$^+$ variants achieve better theoretical convergence rates than P2PCFR$^+$ in normal-form games, a instance of IIGs with only one history for each player, P2PCFR$^+$ consistently significantly outperforms them across all 8 games, with Smooth PCFR$^+$ only performs similar to P2PCFR$^+$ in Goofspiel (4). Furthermore, when compared to three other classic tested CFR algorithms, *i.e.*, vanilla CFR, CFR$^+$, and DCFR, P2PCFR$^+$ consistently significantly outperforms vanilla CFR in almost all games, with CFR$^+$ and DCFR never outperform P2PCFR$^+$.

To evaluate the key motivation behind P2PCFR$^+$—reducing the discrepancy between strategies represented by implicit and explicit accumulated counterfactual regrets to enhance the empirical convergence rate of PCFR$^+$—we present the evolution of this discrepancy over time in both PCFR$^+$ and P2PCFR$^+$. The $\ell_1$-norm quantifies this discrepancy. The results are shown in Figure 2 confirms P2PCFR$^+$'s key motivation. In Leduc Poker, Liar's Dice (5), Liar's Dice (6), and Goofspiel (6), this discrepancy in P2PCFR$^+$ consistently remains smaller than in PCFR$^+$, leading to a faster empirical convergence rate, as illustrated in Figure 1. Additionally, in Liar's Dice (4) and Goofspiel (5), the discrepancy in P2PCFR$^+$ is smaller than in PCFR$^+$ during the first 200 and 500 iterations,

respectively. Subsequently, the difference between the discrepancies of P2PCFR$^+$ and PCFR$^+$ becomes negligible. Since large discrepancies have significant effect on empirical convergence rates as shown in Theorem 4.2, P2PCFR$^+$ continues to converge faster than PCFR$^+$ in these two games. In Kuhn Poker, Liar's Dice (3), and Goofspiel (4), similar to Liar's Dice (4) and Goofspiel (5), the discrepancy in P2PCFR$^+$ is smaller than in PCFR$^+$ during early stages. This phase, however, lasts fewer than 20 iterations, resulting in similar performance between P2PCFR$^+$ and PCFR$^+$.

Finally, we present the empirical convergence rates of P2PCFR$^+$ under various values of $\alpha$. The results are illustrated in Figure 3. Notably, when $\alpha = 0$, P2PCFR$^+$ simplifies to PCFR$^+$. Our observations indicate that when $\alpha$ is relatively small, specifically when $\alpha \leq 10$, P2PCFR$^+$ consistently outperforms PCFR$^+$. However, when $\alpha$ becomes excessively large, for instance at values of 50 or 100, the performance of P2PCFR$^+$ sometimes degrades significantly. This phenomenon is evident in Kuhn Poker, Liar's Dice (4), Goofspiel (4), and Goofspiel (5). We hypothesize that this degradation occurs because, at higher values of $\alpha$, the strategies defined by implicit and explicit accumulated counterfactual regrets in P2PCFR$^+$ become nearly indistinguishable, thereby diminishing the effectiveness of the key insight of PCFR$^+$—looking one step ahead.

## 6 CONCLUSIONS

We propose P2PCFR$^+$, a novel and effective variant of PCFR$^+$. P2PCFR$^+$ employs the misalignment of step sizes in the updates of implicit and explicit accumulated counterfactual regrets to make a more conservative prediction than PCFR$^+$. This conservative prediction reduces the discrepancy between the strategies represented by implicit and explicit accumulated regrets within the same iteration. This reduction in this discrepancy diminishes the significant discrepancy between strategies represented by explicit accumulated counterfactual regrets across two consecutive iterations, which typically slows down the empirical convergence rate of PCFR$^+$. We prove that P2PCFR$^+$ achieves a faster convergence rate compared to PCFR$^+$. Experimental results further validate that P2PCFR$^+$ exhibits a faster empirical convergence rate than PCFR$^+$.

To the best of our knowledge, we are the first to propose the misalignment of step sizes in the updates of implicit and explicit accumulated counterfactual regrets, a simple yet novel technique that effectively improves the empirical convergence rate of PCFR$^+$.

The implementation of P2PCFR$^+$ is straightforward, requiring only a single line modification to PCFR$^+$. Empirically, P2PCFR$^+$ consistently outperforms PCFR$^+$ when $\alpha \leq 10$. This demonstrates that a minor change to PCFR$^+$ can lead to significant performance improvements.

Moreover, the discounting techniques used in DCFR are compatible with P2PCFR$^+$. By incorporating discounting concepts from DCFR, we can assign different values of $\alpha$ to different iterations $t$, further enhancing the empirical convergence rate of P2PCFR$^+$. We consider this as future work.

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

## A    PROOF OF THEOREM 4.3

*Proof.* To prove Theorem 4.1, we use the equivalence between RM$^+$ and Online Mirror Descent (OMD) proposed by Farina et al. (2021).

We first introduce OMD. OMD is a traditional regret minimization algorithm (Nemirovskij & Yudin, 1983). Let $\ell^t \in \mathbb{R}^d$, $x^t \in \mathcal{D}$, and let $\psi : \mathcal{D} \to \mathbb{R}_{\geq 0}^d = \{y | y \in \mathbb{R}^d, y \geq 0\}$ be a 1-strongly convex differentiable regularizer with respect to some norm $\| \cdot \|$, OMD generates the decisions via

$$x^{t+1} := \arg\min_{x' \in \mathcal{D}} \left\{ \langle \ell^t, x' \rangle + \frac{1}{\eta} \mathcal{B}_\psi(x' \parallel x^t) \right\},$$

where $\mathcal{B}_\psi(u, v) = \psi(u) - \psi(v) - \langle \nabla\psi(v), u - v \rangle$ is the Bregman divergence associated with $\psi(\cdot)$.

From the analysis in Section D of Farina et al. (2021), by setting $\psi(\cdot)$ as the the quadratic regularizer $\frac{1}{2} \| \cdot \|_2^2$, the update rule P2PCFR$^+$ at infoset $I$ can be written as

$$
\begin{aligned}
\hat{R}_I^t &\in \arg\min_{R_I \in \mathbb{R}_{\geq 0}^{|A(I)|}} \{\langle -r_I^{t-1}, R_I \rangle + \frac{1}{\eta} \mathcal{B}_\psi(R_I, R_I^t)\}, \\
\hat{R}_I^{t+1} &\in \arg\min_{R_I \in \mathbb{R}_{\geq 0}^{|A(I)|}} \{\langle -(1+\alpha)r_I^t, R_I \rangle + \frac{1}{\eta} \mathcal{B}_\psi(R_I, R_I^t)\},
\end{aligned}
\tag{7}
$$

where $\eta > 0$ is a constant. Note that if $R_I^0 = 0$ for all $I \in \mathcal{I}$, then for any $\eta$, the strategy profile sequence $\{\sigma^1, \sigma^2, \cdots, \sigma^T\}$ generated by P2PCFR$^+$ is same.

**Lemma A.1.** *(Lemma 4 of Farina et al. (2021)) Let $\mathcal{D} \subseteq \mathbb{R}^d$ be closed and convex, let $\ell^t \in \mathbb{R}^d$, $x^t \in \mathcal{D}$, and let $\psi : \mathcal{D} \to \mathbb{R}_{\geq 0}$ be a 1-strongly convex differentiable regularizer with respect to some norm $\| \cdot \|$. Then,*

$$x^{t+1} := \arg\min_{x \in \mathcal{D}} \left\{ \langle \ell^t, x \rangle + \frac{1}{\eta} \mathcal{B}_\psi(x \parallel x^t) \right\}$$

*is well defined (that is, the minimizer exists and is unique), and for all $x' \in \mathcal{D}$ satisfies the inequality*

$$\langle \ell^t, x^{t+1} - x' \rangle \leq \frac{1}{\eta} \left( \mathcal{B}_\psi(x', x^t) - \mathcal{B}_\psi(x', x^{t+1}) - \mathcal{B}_\psi(x^{t+1}, x^t) \right)$$

Considering the second line of Eq. (7), and using Lemma A.1 with $x^t = R_I^t$, $x^{t+1} = R_I^{t+1}$, $x' = R_I$ and $\ell^t = -(1+\alpha)r_I^t$, we have

$$\langle -(1+\alpha)r_I^t, R_I^{t+1} - R_I \rangle \leq \frac{1}{\eta} \left( \mathcal{B}_\psi(R_I, R_I^t) - \mathcal{B}_\psi(R_I, R_I^{t+1}) - \mathcal{B}_\psi(R_I^{t+1}, R_I^t) \right)$$

$$\Leftrightarrow \langle -r_I^t, R_I^{t+1} - \hat{R}_I^t + \hat{R}_I^t - R \rangle \leq \frac{1}{\eta(1+\alpha)} \left( \mathcal{B}_\psi(R_I, R_I^t) - \mathcal{B}_\psi(R_I, R_I^{t+1}) - \mathcal{B}_\psi(R_I^{t+1}, R_I^t) \right).$$

$$\tag{8}$$

Similarly, considering the first line of Eq. (7), and using Lemma A.1 with $x^t = R_I^t$, $x^{t+1} = \hat{R}_I^t$, $x' = R_I^{t+1}$ and $\ell^t = -r_I^{t-1}$, we get

$$\langle -r_I^{t-1}, \hat{R}_I^t - R_I^{t+1} \rangle \leq \frac{1}{\eta} \left( \mathcal{B}_\psi(R_I^{t+1}, R_i^t) - \mathcal{B}_\psi(R_I^{t+1}, \hat{R}_I^t) - \mathcal{B}_\psi(\hat{R}_I^t, R_i^t) \right). \tag{9}$$

Summing up Eq. (8) with Eq. (9), we have

$$\langle -r_I^t, R_I^{t+1} - \hat{R}_I^t + \hat{R}_I^t - R_I \rangle + \langle -r_I^{t-1}, \hat{R}_I^t - R_I^{t+1} \rangle$$

$$\leq \frac{1}{\eta(1+\alpha)} \left( \mathcal{B}_\psi(R_I, R_I^t) - \mathcal{B}_\psi(R_I, R_I^{t+1}) \right) - \frac{1}{\eta(1+\alpha)} \mathcal{B}_\psi(\hat{R}_I^t, R_i^t) + \frac{1}{\eta} (\mathcal{B}_\psi(\hat{R}_I^t, R_i^t) - \mathcal{B}_\psi(R_I^{t+1}, \hat{R}_I^t) - \mathcal{B}_\psi(\hat{R}_I^t, R_i^t))$$

$$\leq \frac{1}{\eta(1+\alpha)} \left( \mathcal{B}_\psi(R_I, R_I^t) - \mathcal{B}_\psi(R_I, R_I^{t+1}) \right) - \frac{1}{\eta(1+\alpha)} \mathcal{B}_\psi(R_I^{t+1}, \hat{R}_I^t) + \frac{1}{\eta} \mathcal{B}_\psi(R_I^{t+1}, \hat{R}_I^t) - \frac{1}{2\eta} \mathcal{B}_\psi(\hat{R}_I^t, R_i^t)$$

$$\leq \frac{1}{\eta(1+\alpha)} \left( \mathcal{B}_\psi(R_I, R_I^t) - \mathcal{B}_\psi(R_I, R_I^{t+1}) \right) - \frac{1}{\eta(1+\alpha)} \mathcal{B}_\psi(R_I^{t+1}, \hat{R}_I^t) + \frac{1}{\eta} \mathcal{B}_\psi(R_I^{t+1}, \hat{R}_I^t) - \frac{1}{2\eta} \mathcal{B}_\psi(\hat{R}_I^t, R_i^t).$$

From $0 \leq \alpha \leq 1$, we have

$$\langle -\boldsymbol{r}_I^t, \boldsymbol{R}_I^{t+1} - \hat{\boldsymbol{R}}_I^t + \hat{\boldsymbol{R}}_I^t - \boldsymbol{R}_I \rangle + \langle -\boldsymbol{r}_I^{t-1}, \hat{\boldsymbol{R}}_I^t - \boldsymbol{R}_I^{t+1} \rangle$$

$$\leq \frac{1}{\eta(1+\alpha)} \left( \mathcal{B}_\psi(\boldsymbol{R}_I, \boldsymbol{R}_I^t) - \mathcal{B}_\psi(\boldsymbol{R}_I, \boldsymbol{R}_I^{t+1}) \right).$$

Arranging the terms, we have

$$\langle -\boldsymbol{r}_I^t, \hat{\boldsymbol{R}}_I^t - \boldsymbol{R}_I \rangle \leq \frac{1}{\eta(1+\alpha)} \left( \mathcal{B}_\psi(\boldsymbol{R}_I, \boldsymbol{R}_I^t) - \mathcal{B}_\psi(\boldsymbol{R}_I, \boldsymbol{R}_I^{t+1}) \right) + \langle \boldsymbol{r}_I^{t-1} - \boldsymbol{r}_I^t, \hat{\boldsymbol{R}}_I^t - \boldsymbol{R}_I^{t+1} \rangle. \tag{10}$$

From the facts that $\boldsymbol{r}_I^t = \langle v^{\sigma^t}(I), \sigma_i^t(I) \rangle \boldsymbol{1} - v^{\sigma^t}(I)$ and $\sigma_i^t(I) = \frac{[\hat{\boldsymbol{R}}_I^t]^+}{\|[\hat{\boldsymbol{R}}_I^t]^+\|_1} = \frac{\hat{\boldsymbol{R}}_I^t}{\|\hat{\boldsymbol{R}}_I^t\|_1}$, we have

$$\langle -\boldsymbol{r}_I^t, \hat{\boldsymbol{R}}_I^t - \boldsymbol{R}_I \rangle = \langle \langle v^{\sigma^t}(I), \sigma_i^t(I) \rangle \boldsymbol{1} - v^{\sigma^t}(I), \boldsymbol{R}_I - \hat{\boldsymbol{R}}_I^t \rangle$$

$$= \langle \langle v^{\sigma^t}(I), \frac{\hat{\boldsymbol{R}}_I^t}{\|\hat{\boldsymbol{R}}_I^t\|_1} \rangle \boldsymbol{1} - v^{\sigma^t}(I), \boldsymbol{R}_I - \hat{\boldsymbol{R}}_I^t \rangle$$

$$= \langle \langle v^{\sigma^t}(I), \frac{\hat{\boldsymbol{R}}_I^t}{\|\hat{\boldsymbol{R}}_I^t\|_1} \rangle \boldsymbol{1} - v^{\sigma^t}(I), \boldsymbol{R}_I \rangle \tag{11}$$

$$= \langle v^{\sigma^t}(I), \sigma_i^t(I) - \boldsymbol{R}_I \rangle.$$

Since $\boldsymbol{R}_I \in \mathbb{R}_{\geq 0}^{|A(I)|}$, we have

$$R^T(I) = \sum_{t=1}^T \sum_{a \in A(I)} \sigma_i^t(Ia) v^{\sigma^t}(Ia) - \min_{a \in A(I)} \sum_{t=1}^T v^{\sigma^t}(Ia) \leq \max_{\boldsymbol{R}_I \in \mathbb{R}_{\geq 0}^{|A(I)|}} \sum_{t=1}^T \langle v^{\sigma^t}(I), \sigma_i^t(I) - \boldsymbol{R}_I \rangle.$$

$$\tag{12}$$

Therefore, from Eq. (10), we can bound

$$\frac{1}{\eta(1+\alpha)} \left( \mathcal{B}_\psi(\boldsymbol{R}_I, \boldsymbol{R}_I^t) - \mathcal{B}_\psi(\boldsymbol{R}_I, \boldsymbol{R}_I^{t+1}) \right) + \langle \boldsymbol{r}_I^{t-1} - \boldsymbol{r}_I^t, \hat{\boldsymbol{R}}_I^t - \boldsymbol{R}_I^{t+1} \rangle \tag{13}$$

to bound $R^T(I)$.

**Lemma A.2.** *(Adapted from Lemma 11 of* Wei et al. (2021)*) Let $\mathcal{D} \subseteq \mathbb{R}^d$ be closed and convex, and suppose that $\psi : \mathcal{D} \to \mathbb{R}_{\geq 0}$ be a 1-strongly convex differentiable regularizer with respect to norm $\|\cdot\|_2$, and let $\mathbf{u}, \mathbf{u}_1, \mathbf{u}_2 \in \mathcal{D}$ be related by the following:*

$$\mathbf{u}_1 = \arg\min_{\mathbf{u}' \in \mathcal{D}} \{ \langle \mathbf{u}', \mathbf{g}_1 \rangle + D_\psi(\mathbf{u}', \mathbf{u}) \},$$

$$\mathbf{u}_2 = \arg\min_{\mathbf{u}' \in \mathcal{D}} \{ \langle \mathbf{u}', \mathbf{g}_2 \rangle + D_\psi(\mathbf{u}', \mathbf{u}) \}.$$

*Then we have*

$$\|\mathbf{u}_1 - \mathbf{u}_2\|_2 \leq \|\mathbf{g}_1 - \mathbf{g}_2\|_2,$$

For Eq. (13), summing up from $t = 1$ to $t = T$, we get

$$\frac{1}{\eta} \mathcal{B}_\psi(\boldsymbol{R}_I, \boldsymbol{R}_I^1) + \sum_{t=1}^T \langle \boldsymbol{r}_I^t - \boldsymbol{r}_I^{t-1}, \boldsymbol{R}_I^t - \hat{\boldsymbol{R}}_I^t \rangle$$

$$\leq \frac{1}{\eta(1+\alpha)} \mathcal{B}_\psi(\boldsymbol{R}_I, \boldsymbol{R}_I^1) + \sum_{t=1}^T \|\boldsymbol{r}_I^t - \boldsymbol{r}_I^{t-1}\|_2 \|\boldsymbol{R}_I^t - \hat{\boldsymbol{R}}_I^t\|_2$$

$$\leq \frac{1}{\eta(1+\alpha)} \mathcal{B}_\psi(\boldsymbol{R}_I, \boldsymbol{R}_I^1) + \eta \sum_{t=1}^T \|\boldsymbol{r}_I^t - \boldsymbol{r}_I^{t-1}\|_2 \|(1+\alpha)\boldsymbol{r}_I^t - \boldsymbol{r}_I^{t-1}\|_2 \tag{14}$$

$$\leq \frac{1}{\eta(1+\alpha)} E + \eta \sum_{t=1}^T \|\boldsymbol{r}_I^t - \boldsymbol{r}_I^{t-1}\|_2^2 ((1+\alpha)\|\boldsymbol{r}_I^t\|_2 + \|\boldsymbol{r}_I^{t-1}\|_2)$$

$$\leq \frac{1}{\eta(1+\alpha)} \mathcal{B}_\psi(\boldsymbol{R}_I, \boldsymbol{R}_I^1) + \eta(2+\alpha)F \sum_{t=1}^T \|\boldsymbol{r}_I^t - \boldsymbol{r}_I^{t-1}\|_2,$$

where the third line comes from Lemma A.2, as well as the last line comes from that $E = \max_{\boldsymbol{R}_I \in \Delta^{|A(I)|}} \frac{1}{2}\|\boldsymbol{R}_I - \boldsymbol{R}_I^1\|_2^2$ and $F = \max_{I \in \mathcal{I}} \|\boldsymbol{r}_I\|_2$.

For the term $\frac{1}{\eta(1+\alpha)}E + \eta(2+\alpha)F\sum_{t=1}^{T}\|\boldsymbol{r}_I^t - \boldsymbol{r}_I^{t-1}\|_2$, since it takes the minimum when $\eta = \sqrt{\frac{E}{(1+\alpha)F\sum_{t=1}^{T}\|\boldsymbol{r}_I^t - \boldsymbol{r}_I^{t-1}\|_2}}$, we have that

$$\frac{1}{\eta(1+\alpha)}E + \eta(2+\alpha)F\sum_{t=1}^{T}\|\boldsymbol{r}_I^t - \boldsymbol{r}_I^{t-1}\|_2 \le 2\frac{EF(2+\alpha)}{1+\alpha}\sum_{t=1}^{T}\frac{1}{2}\|\boldsymbol{r}_I^t - \boldsymbol{r}_I^{t-1}\|_2 \tag{15}$$

Combining Eq. (10), (11), (12), and (16), we have

$$R^T(I) \le 2\sqrt{\frac{2+\alpha}{1+\alpha}EF\sum_{t=1}^{T}\frac{1}{2}\|\boldsymbol{r}_I^t - \boldsymbol{r}_I^{t-1}\|_2}. \tag{16}$$

It completes the proof. □

## B   PROOF OF THEOREM 4.3

*Proof.* From the analysis in Section D of Farina et al. (2021), by setting $\psi(\cdot)$ as the the quadratic regularizer $\frac{1}{2}\|\cdot\|_2^2$, the update rule P2PCFR$^+$ at infoset $I$ can also be written as

$$\hat{\boldsymbol{R}}_I^t \in \operatorname*{arg\,min}_{\boldsymbol{R}_I \in \mathbb{R}_{\ge 0}^{|A(I)|}} \{\langle -\frac{1}{1+\alpha}\boldsymbol{r}_I^{t-1}, \boldsymbol{R}_I\rangle + \frac{1}{\eta}\mathcal{B}_\psi(\boldsymbol{R}_I, \boldsymbol{R}_I^t)\},$$

$$\hat{\boldsymbol{R}}_I^{t+1} \in \operatorname*{arg\,min}_{\boldsymbol{R}_I \in \mathbb{R}_{\ge 0}^{|A(I)|}} \{\langle -\boldsymbol{r}_I^t, \boldsymbol{R}_I\rangle + \frac{1}{\eta}\mathcal{B}_\psi(\boldsymbol{R}_I, \boldsymbol{R}_I^t)\}. \tag{17}$$

**Corollary B.1.** *(Adapted from Corollary 1 5 of Farina et al. (2021)) Let $\psi : \mathcal{D} \to \mathbb{R}_{\ge 0}$ be a 1-strongly convex differentiable regularizer with respect to norm $\|\cdot\|_2$. For all $\hat{x} \in \mathcal{D}$, all $\eta > 0$, and all times $T$, the regret cumulated the algorithm defined in Eq. 17 is bounded as*

$$R^T(I) \le \frac{E}{\eta} + \eta\sum_{t=1}^{T}\|\boldsymbol{r}_I^t - \frac{1}{(1+\alpha)}\boldsymbol{r}_I^{t-1}\|_2^2.$$

From Corollary B.1, we have

$$\begin{aligned}
R^T(I) \le & \frac{E}{\eta} + \eta\sum_{t=1}^{T}\|\boldsymbol{r}_I^t - \frac{1}{1+\alpha}\boldsymbol{r}_I^{t-1}\|_2^2 \\
\le & \frac{E}{\eta} + \eta\sum_{t=1}^{T}\left(\|\boldsymbol{r}_I^t\|_2^2 + \frac{1}{(1+\alpha)^2}\|\frac{1}{1+\alpha}\boldsymbol{r}_I^{t-1}\|_2^2\right) \\
\le & \frac{E}{\eta} + \eta F^2 T\left(1 + \frac{1}{(1+\alpha)^2}\right) \\
\le & 2\sqrt{\left(1 + \frac{1}{(1+\alpha)^2}\right)EF^2T},
\end{aligned}$$

where the third line comes $F = \max_{I \in \mathcal{I}} \|\boldsymbol{r}_I\|_2$, and the last line is from that the third line takes the minimum when $\eta = \sqrt{\frac{E}{F^2 T\left(1 + \frac{1}{(1+\alpha)^2}\right)}}$. It completes the proof. □