# OpenReview forum: "Efficient Predictive Counterfactual Regret Minimization$^+$ Algorithm in Solving Extensive-Form Games"
_ICLR.cc/2025/Conference — Submitted to ICLR 2025_

### Official Review · Reviewer_fqEu · 2024-10-28

**Soundness:** 2
**Presentation:** 3
**Contribution:** 2
**Rating:** 5
**Confidence:** 4

**Summary:**

The authors introduce P2PCFR+, a slight modification of the PCFR+ algorithm for equilibrium computation in extensive-form games. They argue that the instability present in PCFR+ is caused by prediction errors, and leads to worse performance in practice than what is achievable. As a remedy, they propose a method that downweights the prediction by a constant factor, thus reducing its impact. Experiments suggest that their method may converge slightly faster than PCFR+ across a variety of different games.

**Strengths:**

The method introduced by the paper is clean, simple, and (surprisingly, in my opinion) practically effective. Beating PCFR+ in practice is not easy. Experiments are fairly comprehensive, and the paper provides some theoretical guarantees about their method.

**Weaknesses:**

Some worries about the presentation of the theoretical part of the paper prevent me from recommending acceptance in the current state.

The paper makes the claim that it is the predictions that cause PCFR+ to be unstable and thus slow, but this seems hard to support theoretically. The fact that these algorithms are unstable is in some sense a necessary consequence of their scale-invariance, which has been observed to be a *good* property for fast algorithms (see e.g. [1]). So, it is strange to see this paper claiming that this instability is suddenly the source of *bad* performance. Moreover, it is unclear to me that the proposed method resolves this instability (by more than a small approximation): I think for any finite $\alpha$ it should be possible to derive some counterexample in which the strategy jumps from [1, 0] to [0, 1].

The claimed "theoretical improvement" entirely centers around improving the factor $2\sqrt{(2+\alpha)/(1+\alpha)}$ in Theorem 4.1, which is $\sqrt{8}$ for PCFR+ ($\alpha=0$) and $\sqrt{6}$ for $\alpha=1$. This is barely an improvement in theory, and to me does not justify the strong language used by the authors (e.g., "P2PCFR+ exhibits a faster theoretical convergence rate than PCFR+"). Moreover, I am not actually sure that it is an improvement, for example, Theorem 3 of Farina et al (2021) seems to recover the same bound as Theorem 4.1 except with a better constant of $\sqrt{2}$.

The last-iterate convergence of PCFR+ is not known in theory, but in practice it is often very strong, sometimes even better than the linear average. I would recommend that the authors evaluate the last-iterate performance of P2PCFR+ on practical examples as well.

The authors claim that, compared to other recent papers about PCFR+ (e.g., Smooth PCFR+), P2PCFR+ is parameter-free. But this is not true: there is a parameter $\alpha$ to tune! Perhaps more precise is to say that it remains scale-invariant?  The practical improvement over PCFR+ is not that great, and I'm not sure it is enough to justify the additional complexity of introducing the hyperparameter $\alpha$.

Relatively minor issues:
* The example in Section 4.1 is impossible. In that example, we have $R^t_I = [2, 0]$, so $\sigma^t(I)= [1, 0]$. But then the instantaneous counterfactual regret $r^t_I$ must be of the form $[0, \cdot]$, not $[-1, 1]$ as in the paper. I think it is possible to get PCFR+ to go from $[1 ,0]$ to $[0, 1]$, but the construction is a bit tricker, e.g., I think you need to make $R^t_I = [0, 0]$.
* Why "P2PCFR+"? There are only two "P"s in "Pessimistic PCFR+", so shouldn't it be "P2CFR+" or "PPCFR+"?
* In Eq. (6) and Theorem 4.2, writing absolute constants such as $2$ and $(2+\alpha)/(1+\alpha)$ (which is bounded in $[3/2, 2]$ for $\alpha \in [0, 1]$) within a big-O is weird. Can you state those results without any big-Os?

[1] Darshan Chakrabarti, Julien Grand-Clément, Christian Kroer (NeurIPS 2024) "Extensive-Form Game Solving via Blackwell Approachability on Treeplexes"

**Questions:**

1. What makes extending Theorem 4.1 beyond $\alpha=1$ hard? It would be nice to see some results about what happens when $\alpha$ is, say, 5 (as used in the experiments). Also note that $\alpha \to \infty$ just gives plain (non-predictive) CFR+, so the limit $\alpha \to \infty$ can likely also be analyzed. If something interesting can be said for the $\alpha > 1$ case, it may also resolve my concern about theoretical improvement.
2. L297: "Obviously, the value of $\lVert \tilde\sigma_i^{t+1}(I) - \tilde\sigma_i^{t}(I) \rVert$ remains the same as in PCFR+" -- this is not obvious; is it even true? $\tilde\sigma_i^{t+1}(I)$ depends on $R^{t+1}_I$, which depends on $\sigma_i^t(I)$, which is different between P2CFR+ and PCFR+...

---

### Official Review · Reviewer_TSNP · 2024-10-31

**Soundness:** 2
**Presentation:** 2
**Contribution:** 2
**Rating:** 3
**Confidence:** 2

**Summary:**

This paper introduces Pessimistic Predictive CFR that introduces a new parameter $\alpha$ to tune the amount of optimism introduced in the update. In particular $\alpha = \infty$ recovers standard CFR while $\alpha = 0$ recovers the predictive CFR.

The authors show that the constants of the regret bound improve for $\alpha > 0$ justifying some level of pessimism theoretically.

The paper is concluded with some additional experiments.

**Strengths:**

The bounds proved for the new algorithm are stronger for some regime of $\alpha$.

**Weaknesses:**

The improvement shown theoretically is not very evident as it just improves a constant.
Moreover, I have several concerns regarding the proof of Theorem 1.
First of all, I think that the definition of the updates in equation 7 is not consistent with the updates given in the main text. In particular, the first equation just give $\mathbf{R}_t$ and not $\hat{\mathbf{R}_t}$.
Then, when applying Lemma A.1 in Line 684 $x^{t+1}$ should be set equal to

$\hat{\mathbf{R}_{t+1}}$

and not to

$\mathbf{R}_{t+1}$.

All these imprecision makes the reading of the proof very difficult.

Moreover, I think that in the step leading to line 706 there is a mistake indeed taking $\alpha = 1$, we get on the right hand side an additional term equal to $$\frac{1}{2\eta} (\mathcal{B}(\mathbf{R}^{t+1}, \hat{\mathbf{R}}^t) - \mathcal{B}(\hat{\mathbf{R}}^t, \mathbf{R}^{t+1})).$$
The authors end up with the display at line 706 assuming that the above term is negative but why is this the case ?

**Questions:**

Please see the question in weaknesses.

Moreover, given the result in Theorem 1 it seems that the best result for $\alpha = 1$ but this corresponds to basically predict half the current the loss. It does not seem natural to me that predicting half of the current loss gives the best constant. Do you have any explanations for this phenomenon ?

If the authors can rewrite the proof of Theorem 1 without typos / mistakes and answer my questions I am willing to improve my score.

---

### Official Review · Reviewer_UcfV · 2024-11-02

**Soundness:** 2
**Presentation:** 3
**Contribution:** 3
**Rating:** 5
**Confidence:** 3

**Summary:**

The paper proposes P2PCFR+, a variant of PCFR+ that uses smaller step size for computing predicted accumulated regrets compared to accumulated regrets. It decreases the discrepancy between strategies in successive iterations, thereby reducing the error in predicted instantaneous regrets. The proposed algorithm is proved to have a faster convergence rate than PCFR+ and performs well on most testing games.

**Strengths:**

* The proposed method improves performance with only a single line modification to PCFR+.
* The paper is well written and offers a comprehensive overview of backgrounds.
* The experimental results show that the proposed method converges faster than other PCFR+ variants in most games.

**Weaknesses:**

* (Lines 300-303) The discrepancy $\left\Vert \sigma_i^{t+1}(I) - \sigma_i^t(I) \right\Vert$ is not guaranteed to decrease, since the term $\left\Vert \sigma_i^{t+1}(I) - \tilde{\sigma}_i^{t+1}(I) \right\Vert$ may increase.
* (Line 477) DCFR performs well on large-scale poker games such as HUNL subgames. The paper lacks experimental comparisons on these games.
* (Lines 431, 464) There is a gap between theory and experiments in the selection of the hyperparameter $\alpha$. The theorem states that the theoretical convergence rate improves with the increasing of $\alpha$, and therefore the paper deduces that P2PCFR+ converges faster than PCFR+. However, when $\alpha\rightarrow \infty$, the algorithm reduces to CFR+, which performs worse than PCFR+ in most games. Overall, the theory does not fully explain or guide the selection of $\alpha$, and the authors still rely on grid search to identify a suitable hyperparameter.”

**Questions:**

* (Theorems 4.1, 4.3) Why is $\boldsymbol{R}_I \in \Delta^{\left\vert A(I) \right\vert}$ in $E$? Shouldn't it be $\boldsymbol{R}_I \in \mathbb{R}\_{\ge 0}^{|A(I)|}$?
* (Theorem 4.2) Did you forget to divide $T$?
* (Lines 268-269) Why PCFR+ converges more slowly than CFR+? The PCFR+ paper states "We conclude that PCFR+ is significantly faster than CFR+ and DCFR on non-poker games, whereas DCFR is the fastest on poker games".
* (Line 417) Is APCFR+ a typo for PCFR+ or P2PCFR+?
* (Lines 701-703) Can you explain the deduction? The last three terms reduce to $\frac{\alpha}{\eta(1 + \alpha)} \mathcal{B}\_\psi(\boldsymbol{R}_I^{t+1}, \hat{\boldsymbol{R}}_I^t)- \frac{1}{2\eta} \mathcal{B}\_\psi(\hat{\boldsymbol{R}}_I^t, \boldsymbol{R}_i^t)$. Why it can be removed using $0 \le \alpha \le 1$?

---

### Official Review · Reviewer_evJK · 2024-11-04

**Soundness:** 2
**Presentation:** 3
**Contribution:** 2
**Rating:** 3
**Confidence:** 5

**Summary:**

The authors present a variant of CFR (in particular a variant of PCFR+), which performs well on certain classes of two-player zero-sum EFGs.

**Strengths:**

It is interesting that a simple modification to $\textrm{PCFR}^{+}$ yields a generalized algorithm with the same theoretical convergence rates (more on this in the following section) and, on some games, significantly improves empirical performance.

**Weaknesses:**

While the idea is interesting, I am not convinced the contribution meets the threshold for publication, despite occasional superior empirical performance:

Unless I have misunderstood, the improvement in the theoretical convergence rate is in a multiplicative constant, it is misleading to say that the proposed algorithm has a better theoretical convergence rate than PCFR+, and in both cases, the range of the multiplicative constant as $\alpha$ is varied over the interval for which the respective theorems hold (as in Theorem 4.2 and Theorem 4.4) is quite small. Also, by using the language the authors use seems to imply that CFR+ has a "faster" theoretical convergence rate than PCFR+ at least when using the "convergence rates" that are worst-case/agnostic to the discrepancy between prediction and observation(perhaps this is a claim the authors would stand by, given their claim that CFR+ dominates PCFR+ empirically).

Given that the theoretical contributions seem minor, it seems to be perhaps the primary contribution is to provide a method with theoretical guarantees that generalizes PCFR+ with an appropriate choice of $\alpha$. While the experiments demonstrate that the proposed method sometimes significantly outperforms the algorithm (as noted above), the experiments should be run on games beyond Kuhn, Leduc, Liar's Dice and Goofspiel (e.g., it would have been better to show performance on a larger variety of different types of games and include for example Battleship and Pursuit-Evasion; it is noted in the original PCFR+ paper itself that the performance of PCFR+ is not as great on poker/poker-like games). Additionally, see the below question about averaging.

**Questions:**

1. Why not compare using quadratic averaging for PCFR+? The original PCFR+ paper uses quadratic averaging for PCFR+ (and quadratic averaging often leads to significant speedup for PCFR+, while it seems to hurt CFR+); perhaps this would also help the proposed method, and would be worth trying out.
2. In line 416/417 is APCFR+ supposed to be P2PCFR+?

---

### Meta-Review · Area_Chair_vbcx · 2024-12-19

**Metareview:**

The authors proposed a modification of the PCFRP algorithm, with some better theoretical properties and occasional superior performance in practice. All the reviewers agree that the paper has limitations that preclude publication at this time. In particular, there is consensus that the claims regarding theoretical improvements are only with respect to a multiplicative constant and that certain claims of parameter-freeness are not justified given the presence of a parameter $(\alpha)$ that requires tuning. Given this consensus, it was decided to reject the paper.

**Additional Comments On Reviewer Discussion:**

The authors have not posted responses.

---

### Decision · Program_Chairs · 2025-01-22

Reject